# Determination of metformin bio-distribution by LC-MS/MS in mice treated with a clinically relevant paradigm

Kiran Chaudhari[1☯], Jianmei Wang[2☯], Yong Xu[1], Ali Winters[1], Linshu Wang[1], Xiaowei Dong[3], Eric Y. Cheng[3], Ran Liu[1], Shao-Hua Yang[1]*

1 Department of Pharmacology and Neuroscience, Institute for Healthy Aging, University of North Texas Health Science Center, Fort Worth, Texas, United States of America, 2 Pharmaceutical analysis core lab, College of Pharmacy, University of North Texas Health Science Center, Fort Worth, Texas, United States of America, 3 Pharmaceutical Sciences, College of Pharmacy, University of North Texas Health Science Center, Fort Worth, Texas, United States of America

☯ These authors contributed equally to this work.
* shaohua.yang@unthsc.edu

**Data Availability Statement:** All relevant data are within the paper and its Supporting Information files.

**Funding:** This work was partly supported by the National Institutes of Health grants R01NS088596

## Abstract

Metformin, an anti-diabetes drug, has been recently emerging as a potential "anti-aging" intervention based on its reported beneficial actions against aging in preclinical studies. Nonetheless, very few metformin studies using mice have determined metformin concentrations and many effects of metformin have been observed in preclinical studies using doses/concentrations that were not relevant to therapeutic levels in human. We developed a liquid chromatography-tandem mass spectrometry protocol for metformin measurement in plasma, liver, brain, kidney, and muscle of mice. Young adult male and female C57BL/6 mice were voluntarily treated with metformin of 4 mg/ml in drinking water which translated to the maximum dose of 2.5 g/day in humans. A clinically relevant steady-state plasma metformin concentrations were achieved at 7 and 30 days after treatment in male and female mice. Metformin concentrations were slightly higher in muscle than in plasma, while, ~3 and 6-fold higher in the liver and kidney than in plasma, respectively. Low metformin concentration was found in the brain at ~20% of the plasma level. Furthermore, gender difference in steady-state metformin bio-distribution was observed. Our study established steady-state metformin levels in plasma, liver, muscle, kidney, and brain of normoglycemic mice treated with a clinically relevant dose, providing insight into future metformin preclinical studies for potential clinical translation.

## Introduction

Metformin is the first-line drug for the treatment of type 2 diabetes mellitus with a favorable risk/benefit profile. In recent years, metformin has been drawing increasing attention for its potential beneficial effects of cardiovascular protection [1], cancer management [2, 3], counteracting liver lipids accumulation [4], and longevity (NCT 02432287, 01765946, 02745886 at

(SY), R01NS109583 (SY), and American Heart Association Grant 17POST33670981 (KC)

**Competing interests:** The authors have declared that no competing interests exist.

https://clinicaltrials.gov/). Nonetheless, there is emerging preclinical and clinical evidence that not all individuals on metformin will derive the same benefit and some might develop side effects [5]. The very recent MASTERS trial demonstrated that metformin treatment blunts muscle hypertrophy in response to progressive resistance exercise training in older adults, suggesting that favorable effect of metformin on lifespan may not even translate to benefit in all tissues [6].

Metformin's effects on aging and aging-related diseases have been extensively explored in preclinical studies using different doses and concentrations. Most often, the *in vitro* studies use metformin concentrations ranging from µM to mM [7]. Many effects of metformin have been observed in mouse studies using doses that were not relevant to therapeutic levels in human. Most of the studies often fail to establish a direct correlation of the beneficial or detrimental effects with metformin concentrations due to lack of metformin measurement. Our understanding of the mechanisms underlying the action of metformin has dramatically evolved recently, including AMPK activation [8, 9], inhibition of mitochondrial complex I, and mitochondrial glycerophosphate dehydrogenase [10, 11]. There is increasing indication that these actions of metformin might be dose/concentration-dependent [7]. Thus, the outcome parameters after metformin treatment needs to be directly correlated with the dosages as well as actual concentrations of metformin in the organ of interest.

Metformin is widely distributed into different organs, including intestine, liver, skeletal muscle, and brain, and excreted unchanged mainly through kidney [12]. Metformin uptake in different tissues depend on the expression of plasma membrane transporter which could fluctuate between three-to tenfold, leading to dramatic different metformin levels in different organs [13]. Bio-distribution of metformin has been studied in rodents using [11]C-metformin PET imaging in which metformin was found to accumulate in intestine, kidney and liver at much higher concentrations than in plasma after single intravenous administration [14, 15]. Similar PET bio-distribution pattern of [11]C-metformin has also been observed in human after single intravenous or oral administration [16]. Nonetheless, tissue metformin levels are rarely measured in both preclinical and clinical studies. In the present study, we treated C57BL/6J normoglycemic mice with metformin in a clinically relevant paradigm and metformin bio-distribution in the plasma, liver, kidney, muscle, and brain was determined by a LC-MS/MS methods established and verified in the laboratory.

## Materials and methods

### Chemicals and reagents

Metformin hydrochloride (Catalog Number:151691, >98.0% purity) was purchased from MP Biomedicals (Solon, OH). 1,1-Dimethyl-d$_6$-biguanide HCL (Metformin-D6 hydrochloride, CAS:1185166-01-1, 98.5% purity) was purchased from CDN Isotopes (Pointe-Claire, Quebec, Canada). Methanol Optima™ (Catalog Number: A456), Acetonitrile Optima™ (Catalog Number: A955), Formic Acid Optima™ (Catalog Number: A117) and Ammonium Acetate Optima™ (Catalog Number: A11450) were LC-MS grade and purchased from Fisher Scientific (Pittsburgh, PA). Ultra-pure water was obtained from a Milli-Q Plus water purification system (Millipore, Bedford, MA).

### Animals and metformin treatment

Procedures for animal treatment were approved by the University of North Texas Health Science Center Institutional Animal Care and Use Committee. C57BL/6J mice (male and female, 2.5-months old) were purchased from the Jackson Laboratory (Bar Harbor, ME), housed singly in clear polycarbonate cages at 23 ± 1°C under a 12-hour light/dark cycle, and fed *ad*

*libitum*. Total 24 mice including 12 male and 12 female mice were used for this project. Metformin hydrochloride (Sigma-Aldrich, St. Louis, MO) was added to drinking water (4 mg/ml), and mice were allowed to drink *ad libitum*. Water intake and body weights were measured during the 5-days after treatment. At 7 or 30 days after treatment, water intake was withheld for 4 hours, and then mice were anesthetized using isoflurane inhalation anesthesia, and blood was collected in EDTA tubes via cardiac puncture. Isoflurane (#NDC-66794-017-25; Piramal Healthcare, Boston, MA, USA) anesthesia induction was conducted in the open drop chamber. Further, during cardiac puncture anesthesia was maintained using a 15 ml tube nose cone. The procedure of cardiac puncture was carried out without opening the chest using 22-gauge needle attached to 1 ml syringe. The needle was inserted just below left end of sternum, pushing horizontally, towards the heart until blood is noticed in the syringe. The cardiac puncture and blood collection procedure was completed in less than one minute under maintenance anesthesia. The tubes were centrifuged at 10,000 X g for 15 min, and supernatant plasma was collected and snap-frozen in liquid nitrogen. Immediately after cardiac puncture, the mouse was euthanized with cervical dislocation followed by perfusion with cold normal saline to remove remaining blood from the organs. The liver, kidney, brain, and muscle were collected and snap-frozen in liquid nitrogen. All samples were stored in -80 ˚C freezer until further assessment.

## Sample preparation

Since metformin does not bind to proteins, a single-step protein-precipitation extraction procedure was adopted to extract metformin from mouse tissues and plasma. All standards and Quality control samples, and study samples processed with the same procedure as followings. All metformin-treated tissues (liver, brain, kidney, muscle) and blank tissues for preparation of standard curve and quality control samples were homogenized in 1:15 Milli-Q water using GLH-01 homogenizer and Sonic Ruptor 250 ultrasonic homogenizer (OMNI international the homogenizer company, Kennesaw, GA). To the 40 μl of mouse tissue homogenate or 20 μl of mouse plasma, 50 μl of metformin-D6 (250 ng/ml) was added, and vortexed for 2 minutes, then 300 μl of acetonitrile was added and vortexed for 5 minutes. This mixture was then centrifuged at 16,000 X g for 10 minutes. The supernatant was transferred into 1.9 ml glass autosampler vials (catalog number:03-391-8, 03-391-9, Fisher brand, Fisher Scientific, Pittsburgh, PA) before adding 1.0 ml of mobile phase A (2 mM Ammonium acetate in water). An aliquot of 5 μl of the diluted supernatant was directly injected on LC-MS/MS.

## Liquid chromatographic and instrumentation conditions

LC-MS/MS system was performed using an Agilent 1260 Infinity HPLC and HiP ALS (Autosampler) coupled with Agilent 6460 triple quadrupole mass spectrometer with a Jet Stream electrospray ionization source (Agilent Technologies, Santa Clara, CA). All data were acquired employing Agilent Mass Hunter software 7.0. Separation of Metformin from tissue or plasma samples was achieved at ambient temperature on Waters XBridge C18 column (3.0 x 50 mm, 3.5 μm). The mobile phases consist of 2 mM ammonium acetate buffer in water as mobile phase A and 100% acetonitrile as mobile phase B in a gradient run at a flow rate of 0.35 ml/min. The run started initially at 5% mobile phase B for 1.0 minute, then from 5% to 95% mobile phase B for 1.0 minute, and 95% mobile phase B for 1.0 minute and post-run staying at 5% for 1.0 minute. The total run time is 3.0 minutes, post-run 1.0 minute, and the MS scan window was set between 0.5–1.5 minutes. The mass spectrometer was operated in a positive mode using multiple reaction morning (MRM) with an ion spray voltage at +3.5 kV, the gas

temperature at 325 ˚C, drying gas at flow-rate of 8 Lmin$^{-1}$, nozzle voltage 500 V. The optimized MRM, fragmentor, and collision energy parameters are presented in S1 Table.

## Preparation of calibration curve and quality control samples

All blank tissues or metformin-free tissues (liver, brain, kidney, muscle) for preparation of standard curve and quality control samples were homogenized in 1:15 Milli-Q water using GLH-01 homogenizer and Sonic Ruptor 250 ultrasonic homogenizer (OMNI international the homogenizer company, Kennesaw, GA). All blank plasma or metformin-free plasma were ready for preparation of calibration standards and quality control samples.

A stock solution of metformin was prepared in 50% methanol-water to give a final concentration of 120 μg/ml. The solution was then serially diluted with 50% methanol-water to obtain standard working solutions over a concentration range of 0.2–120 μg/ml. All solutions were stored at -20 ˚C. Calibration standard solutions were prepared by spiking 190 μl of a blank mouse tissue homogenate or blank mouse plasma with 10 μl of a metformin standard solution (ratio 19:1) to give a metformin concentration range of 10–6,000 ng/ml. The quality control (QC) samples, which were used both in the validation study and during each experimental run of the tissue distribution study, were prepared in the same manner as the standard calibration samples.

To the 40 μl of calibration standards and quality control samples in tissue homogenate or 20 μl of calibration standards and quality control plasma samples, 50 μl of metformin-D6 (250 ng/ml) was added, and vortexed for 2 minutes, then 300 μl of acetonitrile was added and vortexed for 5 minutes. This mixture was then centrifuged at 16,000 X g for 10 minutes. The supernatant was transferred into 1.9 ml glass autosampler vials (catalog number: 03-391-8, 03-391-9, Fisher brand, Fisher Scientific, Pittsburgh, PA) before adding 1.0 ml of mobile phase A (2 mM Ammonium acetate in water). An aliquot of 5 μl of the diluted supernatant was directly injected on LC-MS/MS.

## Results and discussion

### Optimization of LC-MS/MS conditions

Electrospray ionization (ESI) was selected for the ionization source of the present LC-MS/MS study because it provided strong signal intensity for metformin and metformin-D6. The most abundant ion in the product ion mass spectrum was m/z 60.2 for metformin. MS conditions, including capillary temperature, spray voltage, source CID, and collision pressure, did not significantly influence the MS behavior of the metformin and were maintained at the auto-tuned values. We adopted two ion pairs in monitoring metformin and used metformin-D6 to improve the precision of the analytical method. It is crucial to have good resolution and separation in chromatography, together with the use of quantifiers and qualifiers to confirm metformin and potentially existed interferences. So two MRM transitions for metformin were monitored to provide sufficient identification of metformin: the quantifier was used for all validation parameters and the qualifier was used for the confirmatory analysis of metformin [17]. The most prominent precursor–product transition ion m/z 60.2 was used as quantifiers for quantification; the second most abundant transition ions m/z 71.2 were used as qualifiers to confirm the presence of metformin in the samples (Fig 1). By monitoring the ion ratio for tissue and plasma samples, we are able to identify any unexpected interference or co-eluted peaks, which could cause the ion ratio to fail.

There is much interest in structural studies of metformin tautomerization [18]. Tautomerization is exploring the relationships between the structure and molecular mechanism of metformin's extraordinarily diverse biological activities. Tautomerism in bioactive compounds

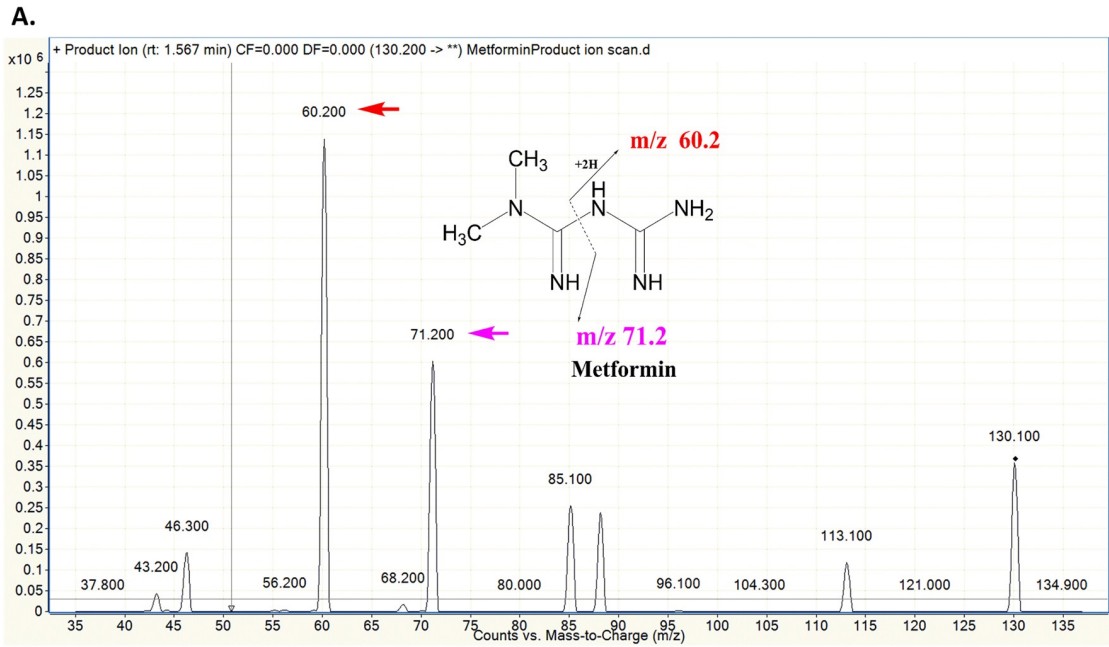

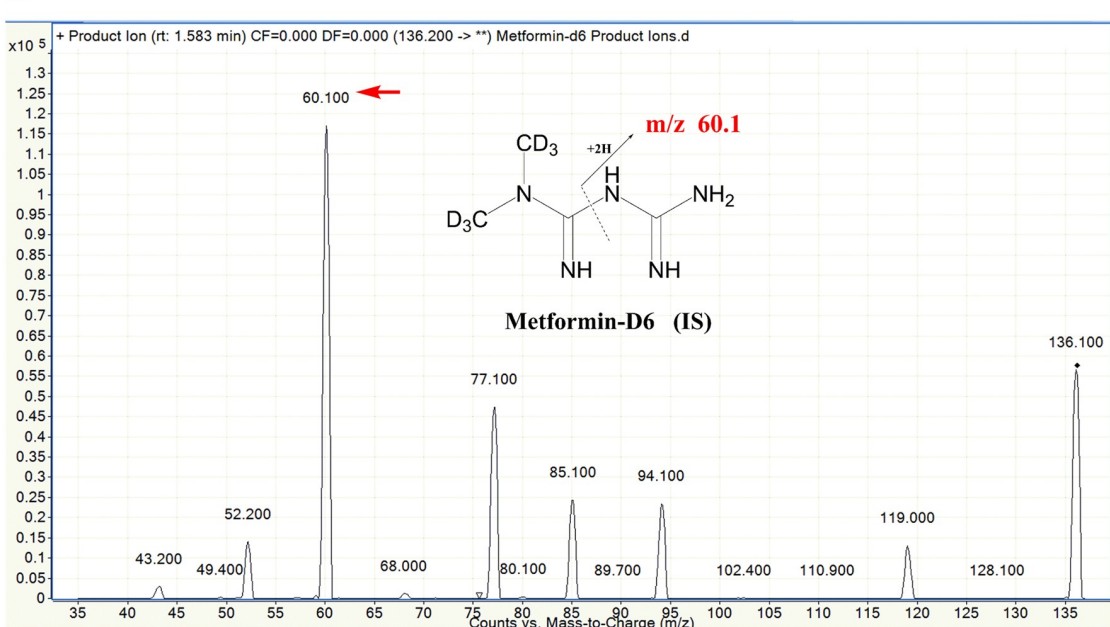

**Fig 1.** A. Metformin and B. metformin-D6 fragmentation ion scans. Y-axis is Relative intensity (cps); X-axis is mass-to-charge (m/z). Red arrow indicates peak for m/z 60.2, pink arrow indicates peak for m/z 71.2.

plays a crucial role in the orientation of bioactivity of drugs that have found wide application in drug design. In order to understand the mechanism of action, tautomeric representations of drug molecules need to be in their most appropriate form. X-ray crystal structure analysis and various spectroscopic studies such as UV,1H, and15N NMR have confirmed that metformin as one of the biguanides exists as tautomer [19–21].

In the initial sample preparation, we applied acetonitrile to precipitate protein in homogenate and plasma samples, and then dilute supernatant with 0.1% formic acid in water. We observed the distorted peak shape in all tissue extracts but not in plasma extract. Decreasing or increasing the composition of the mobile phase dilution solution had no improvement to the peak shape. We monitored void volume during method development. The extracted samples using protein precipitation method typically contained large amount of organic solvent, which is not compatible with the initial 95% of aqueous mobile phase composition. Hence, we diluted the extract with large volume of mobile phase A to keep highly polar metformin in aqueous solution. To avoid metformin elute in solvent front in void area around 0.7min, we applied high resolution C18 column and 95% 2 mM ammonium acetate solution for 1.0 minute as post-run and initial run to re-equilibrate the column, and applied 95% acetonitrile for 1.0 minute to elute other non-polar interferences or co-eluting compounds if there were any. Re-equilibrium of the column for 1.0 minute in post-run is able to give consistent metformin chromatography at retention time 0.89 min, well separated from void area.

During sample preparation, we initially tried 0.1% HCOOH as dilution solution; we observed random peak split for metformin in study samples. Metformin-D6 also exhibited similar peak split or distortion. Previous reports indicated, majority methods used ammonium formate or ammonium acetate solution, adjusted the pH with formic acid as mobile phase (discussed later in Table 4). In pursuit of symmetric peak shape as well as retention time away from solvent front, we decided to use ammonium acetate not only as dilution solution but also as the mobile phase in sample preparation. This adjustment helped pushing the run further away from the solvent front. Initially, we tried 10 mM ammonium acetate as dilution solution and mobile phase A. Although, the LC system built up salt quickly decreasing the MS signal when injecting multiple samples over time, yet, we always got uniform and sharp peak shapes. Then we narrowed down to enough volume of 2 mM ammonium acetate as dilution solution and mobile phase A obtaining uniform and sharp peaks without compromising instrument performance. Hence, we decided to use 2 mM ammonium acetate as mobile phase and dilution solution. It is worth noting that adjustment of pH with formic acid in ammonium acetate solution could give similar results. In our study, we were able to obtain satisfied signals and good chromatography for the study without a need for pH adjustment.

To further troubleshoot the peak distortion and split observed with both metformin and metformin-D6; we used HPLC-TOF-MS and confirmed that the metformin itself underwent split or tautomerism. The peak distortion has been reported in several papers regardless of the column or mobile phase used [22, 23]. The peak distortion could cause peaks merging into void peak area, and that is also the reason we strived to improve peak shape in method development.

We predicted that tautomerization could occur during the sample preparation, and 2 mM ammonium acetate buffer was the best option to control the pH of metformin extract as metformin pH in water was around 6.68 (pKa 12.4) which yielded single and symmetric peak shapes and reproducible and sensitive signals in all tissue extracts. The 2 mM ammonium acetate in the water at 5% composition as initial mobile phase composition gave metformin and metformin-D6 at retention time 0.89 minutes and was reproducible in the entire run (Fig 2).

## Method validation

All data were processed using Mass Hunter Quantitative analysis software version B.07.00 (Agilent Technologies, Santa Clara, CA). The validation of this procedure was performed in order to evaluate the method in terms of selectivity, carryover assessment, the linearity of response, accuracy, precision, and matrix effect according to the US Food and Drug

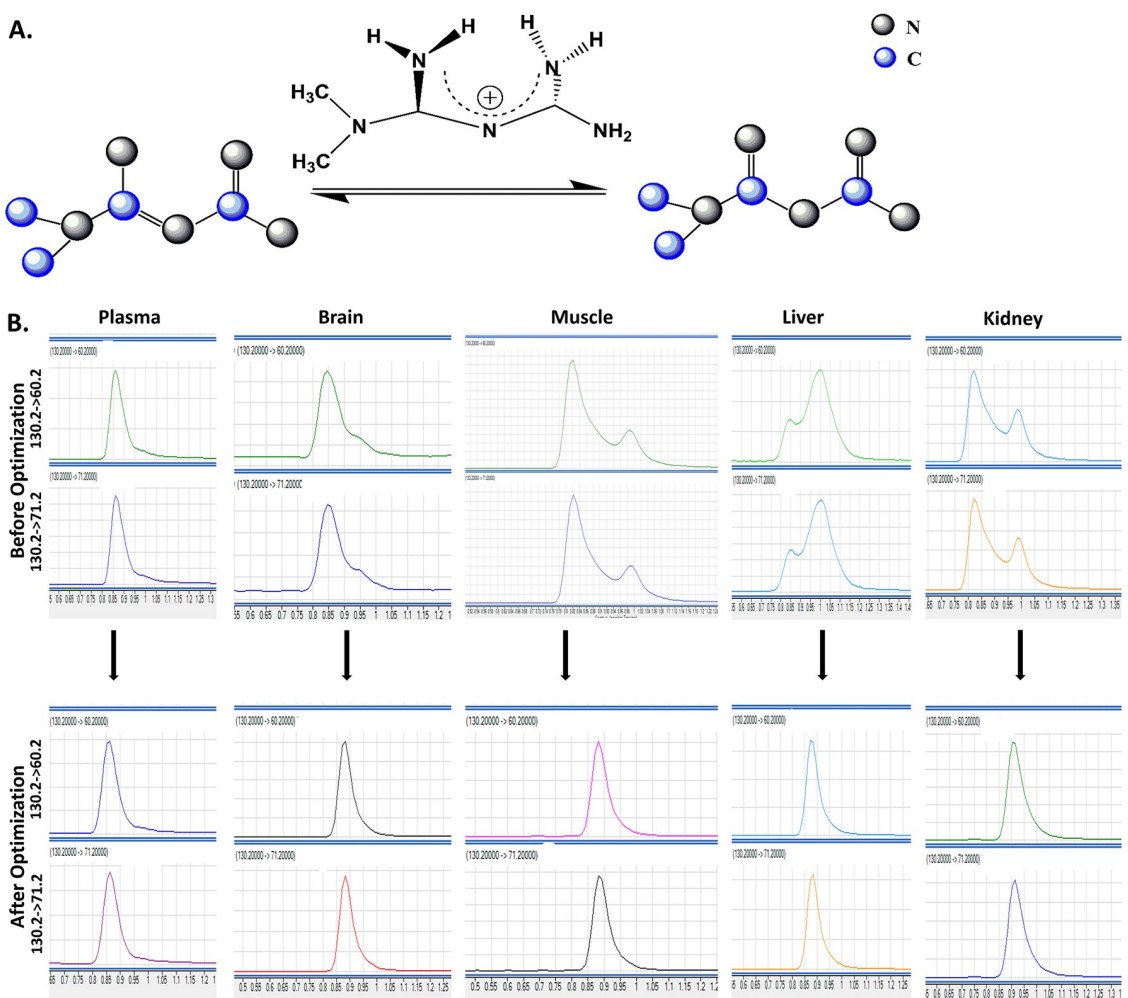

**Fig 2.** Variation of Metformin chromatography by A. Metformin tautomerization in terms of proton movement; B. Representative Chromatographs before and after optimization from left to right: plasma, brain, muscle, liver, and kidney. For each graph, Y-axis: Intensity abundance (cps) and X-axis: acquisition time (min).

Administration Guidelines for Industry using six sets of spiked plasma and tissue QC samples at lowest limit of quantitation concentration (LLOQ, 10 ng/ml), low (LQC, 50 ng/ml), medium (MQC, 1,000 ng/ml), and high (HQC, 3,000 ng/ml) concentrations on the same day and over three days [24]. In MRM mode, blank plasma samples and tissue samples from different mice showed no interference from endogenous tissue or plasma. The typical chromatography of five different blank matrixes (Plasma, Brain, Muscle, Liver, and Kidney) injected right after the upper limit of quantitation standard, together with no matrix solvent as well as those spiked with metformin at the LLOQ (10 ng/ml) are shown in Fig 3. Carryover assessment was evaluated by injecting a double blank following the upper limit of quantitation standard (ULOQ, 6,000 ng/ml) and comparing the signal of this blank with the signal of the preceding LLOQ standard. No significant carryover or contamination was detected within the calibration range.

The calibration curve was linear over the concentration range of 10–6,000 ng/ml for metformin. After comparing the two weighting models ($1/x$ and $1/x^2$), a regression equation with a weighting factor of $1/x$ of the drug to the IS concentration was found to produce the best fit

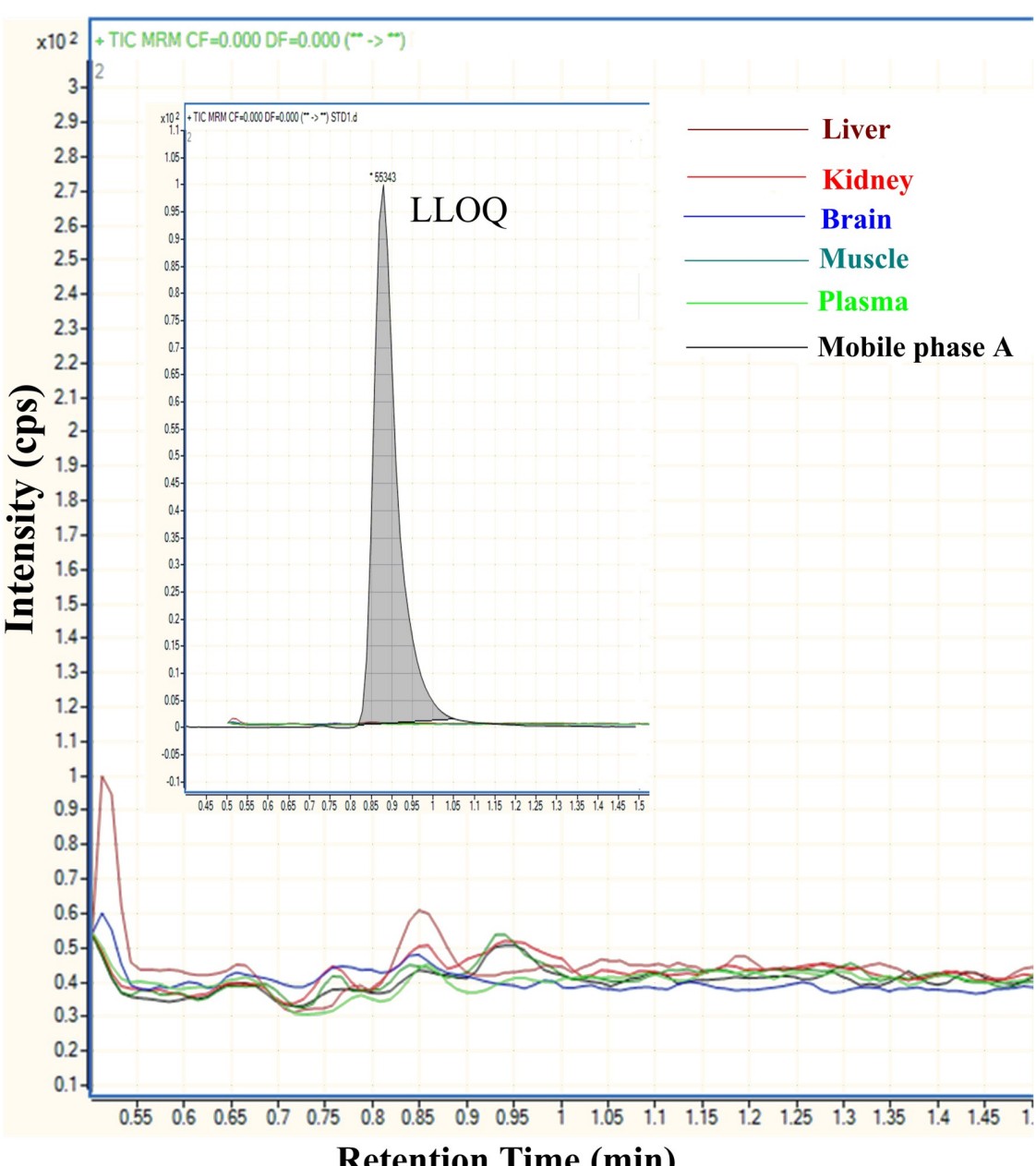

**Fig 3. Representative MRM LC-MS/MS overlaid chromatogram of metformin at a lower limit of quantitation concentration and representative individual matrix blank injection after ULOQ.** Each individual tissue was represented by colors- green-plasma; blue-brain; dark green-muscle; brown-liver; red-kidney; black-mobile phase A. Y-axis: Intensity abundance (cps) and X-axis: acquisition time (min).

for the concentration–detector response relationship for all tissues and plasma. The mean correlation coefficient of the weighted calibration curves generated during the validation was ≥ 0.995 (S2 Table and S1 Fig).

The results for intra-day and inter-day precision and accuracy in plasma and tissue quality control samples are summarized in Table 1. The results supported that the method was accurate with an excellent accuracy range of 94.4–104.3%, and the values of CV were all within

**Table 1. Intra-day and inter-day precision and accuracy of the method in plasma and tissue homogenate spiked with metformin.**

| Sample | Theoretical Concentration (ng/mL) | Intraday (n = 6) | | | Inter-days (n = 18) | | |
|---|---|---|---|---|---|---|---|
| | | Mean (ng/ml) | Precision[a] (% CV) | Accuracy[b] (% bias) | Mean (ng/ml) | Precision[a] (% CV) | Accuracy[b] (% bias) |
| **Plasma** | 10 | 11.9 | 13.87 | 118.9 | 10.83 | 12.96 | 108.33 |
| | 50 | 51.68 | 8.28 | 103.36 | 49.08 | 6.05 | 98.16 |
| | 1,000 | 1,043.25 | 14.16 | 104.32 | 1019.92 | 10.66 | 101.99 |
| | 3,000 | 3,056.15 | 6.48 | 101.87 | 3182.41 | 6.68 | 106.08 |
| **Liver** | 10 | 8.44 | 12.34 | 84.4 | 8.48 | 1.25 | 84.8 |
| | 50 | 52.04 | 3.87 | 104.08 | 48.56 | 6.3 | 97.13 |
| | 1,000 | 948.1 | 2.98 | 94.81 | 961.16 | 3.47 | 96.12 |
| | 3,000 | 2,833.16 | 1.03 | 94.44 | 2847.56 | 5.94 | 94.92 |
| **Brain** | 10 | 11.80 | 12.67 | 118.00 | 10.78 | 18.54 | 107.8 |
| | 50 | 52.48 | 10.79 | 104.96 | 50.91 | 8.00 | 101.81 |
| | 1,000 | 945.78 | 1.72 | 94.58 | 987.43 | 1.61 | 98.74 |
| | 3,000 | 3,013.69 | 6.65 | 100.46 | 3201.4 | 4.87 | 106.71 |
| **Kidney** | 10 | 11.26 | 10.5 | 112.6 | 10.64 | 15.27 | 106.4 |
| | 50 | 47.26 | 3.03 | 94.53 | 54.84 | 4.4 | 109.67 |
| | 1,000 | 976.42 | 2.98 | 97.64 | 1002.2 | 0.82 | 100.22 |
| | 3,000 | 2,854.30 | 2.06 | 95.14 | 2889.02 | 1.04 | 96.3 |
| **Muscle** | 10 | 10.70 | 15.67 | 107.1 | 11.25 | 14.32 | 112.5 |
| | 50 | 48.89 | 2.44 | 97.79 | 45.7 | 3.43 | 91.4 |
| | 1,000 | 1,000.79 | 2.51 | 100.08 | 1005.76 | 2.02 | 100.58 |
| | 3,000 | 3,021.40 | 6.89 | 100.71 | 3154.9 | 5.67 | 105.16 |

[a]: % coefficient of variation (%CV).

[b]: % difference between the average value and its theoretical value.

LLOQ = 10 ng/ml, LQC = 50 ng/ml, MQC = 1000 ng/ml, HQC = 3000 ng/ml.

acceptance criteria of $\leq$ 20% at LOQ level and $\leq$ 15% for the other concentrations. Overall, all experiment accuracy values were within the acceptable range of 100 ± 20% at all concentrations.

The recovery of metformin was determined at three different QC levels (LQC, MQC, and HQC) by comparing the peak area of metformin from extracted QC samples (n = 6 for each level) with the peak area of post-extracted blank plasma spiked with metformin at the corresponding concentration. Recovery was measured by following equation,

$$\text{Recovery} = (\text{Peak area of metformin in QC sample}/\text{Peak area of metformin at same}$$
$$\text{concentration in post} - \text{extracted sample}) \text{ x } 100\%$$

The effect of matrix constituents over the ionization of analytes and IS was determined by comparing the peak area ratios (metformin/metformin-D6) of extracted plasma QC samples with the peak area ratios (metformin/metformin-D6) of the solution prepared in deionized water at corresponding concentrations. The calculated Relative Standard Deviation percentage (RSD%) variation limit was expected to be less than 15%. This determination was performed at three different concentration levels (LQC, MQC, and HQC, n = 6 each).

$$\text{MF}\% = (\text{Peak area ratio of analyte}/\text{IS in extracted samples at QC concentration levels}/\text{peak area}$$
$$\text{ratio of analyte}/\text{IS prepared in deionized water at QC concentration levels}) \text{ x } 100\%$$

**Table 2. Matrix effect (% MF) and recovery (%) of QC samples (n = 6) for metformin in spiked mouse plasma.**

| QC level | MF(%) | RSD% | Recovery (%) | RSD% |
|---|---|---|---|---|
| LQC | 91.20% | 2.33% | 112.80% | 9.87% |
| MQC | 103.32% | 2.53% | 97.90% | 6.12% |
| HQC | 112.97% | 6.81% | 96.60% | 9.54% |

RSD% = Percentage relative standard deviation.

In our study, the matrix effect was evaluated by analyzing three levels of QC samples (LQC, MQC, and HQC) for metformin. The matrix effect for plasma was calculated and lay within the range of 91.2% to 112.9%. The average matrix effect values obtained were < 10% RSD. The extraction recoveries at three concentration levels ranged from 96.6% to 112.8%. The results indicated that the protein precipitation method showed a high recovery for metformin in Table 2. No significant peak-area differences were observed. By applying isotope internal standards metformin-D6 for quantification in large diluted volume, the matrix effect showed little effect on the quantitative analysis of metformin. This demonstrated that the method is robust for metformin and free from interference from the matrix. Therefore, the single-step protein precipitation procedure and extraction of metformin used in this analytical method was efficient and simple.

The observation of two MRM transitions, indicating the chromatographic peak of the analyte at the expected retention time and the resulting area ratio (ion ratio) is considered robust verification criteria (<15%). Expected ion ratios were calculated by averaging the ion ratios of the standards used for the calibration curve. Sample ion ratio can deviate from the failures related to integration discrepancy, or signal loss less than LOD or saturation from a single (or both) MRM transitions in the detector or an interference peak present at the same retention time as that of the analyte. The ion ratio-matching percentile is to compare the sample ion ratio to the average ion ratio of the standard curve. In muscle samples, the ion ratio was found at 92.27 ± 0.45 (Mean ± S.D.). The matching percentile for each sample is from 98.5–100.4%.

The analyte proved to be stable in conditions likely to be present during sample collection, storage, and processing (S3 Table). All data is shown as mean% of nominal value ± S.D. Stock and working stocks were stable in the freezer at -20 ˚C for 30 days. The long-term stability for plasma at -80 ˚C was evaluated and stable for 82 days. Plasma samples can be safely stored at room temperature on a benchtop up to 8 hours (92.1% to 95.8%). Extracted plasma and tissue samples were stable at room temperature up to 48 hours (96.3% to 108.7%). Freeze-thaw stability studies showed the analyte stable for two freeze-thaw cycles (89.7% to 102.5%), and the long-term stability studies showed the analyte stable for 82 days at -80 ˚C (95.6% to 112.5%). Tissue homogenate was analyzed immediately after homogenization.

## Bio-distribution of metformin in normoglycemic mice treated with a clinically relevant paradigm

In preclinical studies, the effects of metformin treatment on aging and aging-related disorders have been extensively explored in mice with different treatment paradigms, while plasma metformin concentrations have been rarely determined. For preclinical studies, it is often convenient and less interruptive to the animals to add medication in drinking water or diet. After adjusting for body weight, the calculated water intake as ml/kg body weight/day was similar in males and females. The average body weight adjusted metformin intake was 534.99 +/- 22.12, 518.28 +/- 23.34, 496.05 +/- 14.46, and 518.25 +/- 26.30 mg/kg body weight/day in male 7-day,

**Table 3. Metformin concentrations in plasma, brain, liver, kidney, and muscle after 7 or 30 days of metformin treatment in male and female mice.**

| Gender | | Male | | Female | |
|---|---|---|---|---|---|
| **Duration of Metformin treatment** | | 7 days | 30 days | 7 days | 30 days |
| **Metformin Intake (mg/kg/day) Calculated Human (60 kg) Equivalent dose (mg/day)** | | 534.99 ± 22.12 <br> 2,602.69 ± 107.60 | 518.28 ± 23.34 <br> 2,521.46 ± 113.55 | 496.05 ± 14.46 <br> 2,412.97 ± 70.36 | 518.25 ± 26.30 <br> 2,521.46 ± 127.95 |
| **Metformin concentration** | **Plasma**(ng/ml) ($\mu M$) | 2,385 ± 181.1 *(18.46 ± 1.40)* | 2,926 ± 283.4 [@] *(22.65 ± 2.19)* | 1,679 ± 242.0 *(13.00 ± 1.87)* | 2,853 ± 275.8 [*, ¶] *(22.09 ± 2.14)* |
| | **Brain** (ng/g) | 438.3 ± 38.58 | 509.9 ± 18.17 | 431.2 ± 24.98 | 675.1 ± 55.91 [$] |
| | **Liver** (ng/g) | 7,993 ± 1,603 | *7,359 ± 589* | 6,191 ± 1,754 | 11,129 ± 2,021 |
| | **Kidney** (ng/g) | 12,636 ± 1,964 | *14,350 ± 1,521* | 11,150 ± 1,860 | 18,372 ± 1,490 [#] |
| | **Muscle** (ng/g) | 3,282 ± 478.5 | 2,923 ± 122.4 | 3,452 ± 336.1 | 4,532 ± 471.7 [&] |

[*] $p<0.05$, female 30 days plasma vs female 7 days plasma;

[$] $p<0.05$, female 30 days brain vs female 7 days brain and male 30 days brain;

[#] $p<0.05$, female 30 days kidney vs female 7 days kidney;

[&] $p<0.05$, female 30 days vs male 30 days muscle.

[@] $p<0.01$, male 30 days plasma vs male 30 days brain, liver, kidney;

[¶] $p<0.01$, female 30 days plasma vs female 30 days brain, liver, kidney. n = 6 each group; metformin molecular weight = 129.16 was used to convert plasma ng/ml to μM.

male 30-day, female 7-day, and female 30-day treatment group, respectively (Table 3). The human equivalent dose was calculated using mice $K_m = 3$ and human $K_m = 37$ as described previously [25]. The 4 mg/ml metformin treatment was equivalent to ~2.5 g /day dose for a 60 kg human. Consistently, our LC-MS/MS metformin analysis demonstrated that metformin 4 mg/ml in water *ad libitum* treatment paradigm yielded plasma concentrations in the range of therapeutic level in patients subjected to maximum dose treatment (Table 3).

The pharmacokinetics of metformin has been well determined in humans. Nonetheless, many clinical studies determined plasma metformin concentrations with different doses and mixed genders [26]. Oral doses of 500 to 1,000 mg of immediate-release metformin are rapidly absorbed and typically yield a peak plasma concentration of 2,000 ng/ml and rarely > 4,000 ng/ml, with a steady-state concentration range of 300 to 1,500 ng/ml [27]. Average steady plasma metformin concentrations have been found at 1,340 and 1,350 ng/ml after 2,500 mg daily dose in healthy subjects and diabetic patients, respectively [12, 28]. In 95 diabetic patients, irrespective of gender, dose, and regularity of intakes, metformin serum concentrations averaged 1,846 ng/ml [29]. In a total of 798 plasma samples from 467 male and female patients, metformin concentrations have been found to be ~2,700 ng/ml [30]. Metformin plasma levels higher than 5,000 ng/ml (38.71 μM) are generally found when metformin is implicated as the cause of lactic acidosis [31]. In our treatment paradigm, steady plasma metformin concentrations were 2,926 ± 283.4 ng/ml and 2,853 ± 275.8 ng/ml at 30 days after treatment in male and female mice, respectively, which is within the range of plasma level in patients subjected to maximum dose treatment.

Interestingly, treatment metformin in diet seems have much higher bioavailability than metformin treatment in drinking water. Metformin treatments of 0.1% and 1% w/w in diet yielded serum concentrations of 0.45 and 5 mM, respectively, which are considerably higher than the plasma concentrations seen in humans treated with metformin [32]. Even intermittent treatment of 1% metformin in diet given every-other-week or 2 consecutive weeks per month produced serum metformin concentration many folds higher than the recommendation of maximum plasma metformin concentration of 2,500 ng/ml (~19 μM) from the considerations of lactic acidosis [12, 33].

Metformin is widely distributed into different organs, including intestine, liver, muscle kidney, and brain, by organic cation transporters (OCTs), and excreted unchanged mainly in the urine [12]. A similar pattern of metformin biodistribution has been observed in rodents and humans with higher metformin accumulation in kidney, intestine, and liver than in plasma using [11]C-metformin PET imaging [14–16]. The liver-to-arterial blood and kidney-to-arterial blood ratios were approximately 5 and 8, respectively, after oral [11]C-metformin administration in healthy human while skeletal muscle [11]C-metformin activity only slightly exceeded that in plasma [16]. We observed that steady metformin concentrations were ~3 and 6 folds higher in the liver and kidney than in plasma ($p < 0.01$), respectively, after 30-day metformin administration at 4 mg/ml in drinking water. Similarly, slightly high metformin concentrations were found in skeletal muscle than in plasma although no significant difference was reached. In addition, concentration of metformin was low in the brain at less than 20% of the plasma level ($p < 0.01$) (Table 3). Often plasma levels of drugs are used as drug monitoring, assuming that the plasma level reflects the accumulation of the drug in various tissues. When analyzed the data for linear regression between plasma and each of tissues, we observed that plasma metformin concentration was significantly correlated with liver ($r^2 = 0.339$, $p = 0.0028$) and brain ($r^2 = 0.4115$, $p = 0.0007$) but not with kidney ($r^2 = 0.08257$, $p = 0.2066$) or muscle ($r^2 = 0.1708$, $p = 0.0559$) (Fig 4).

Most of the clinical studies with plasma metformin measurement did not separate males and females with few exceptions. After oral administration of 850 mg metformin, maximum plasma metformin concentration reached 1,208 and 1,341 ng/ml in healthy male and female subjects, respectively [34]. In type 2 diabetic patients, plasma steady-state metformin concentration reached 1,698 and 1,845 ng/ml in male and female subjects, respectively, after multiple

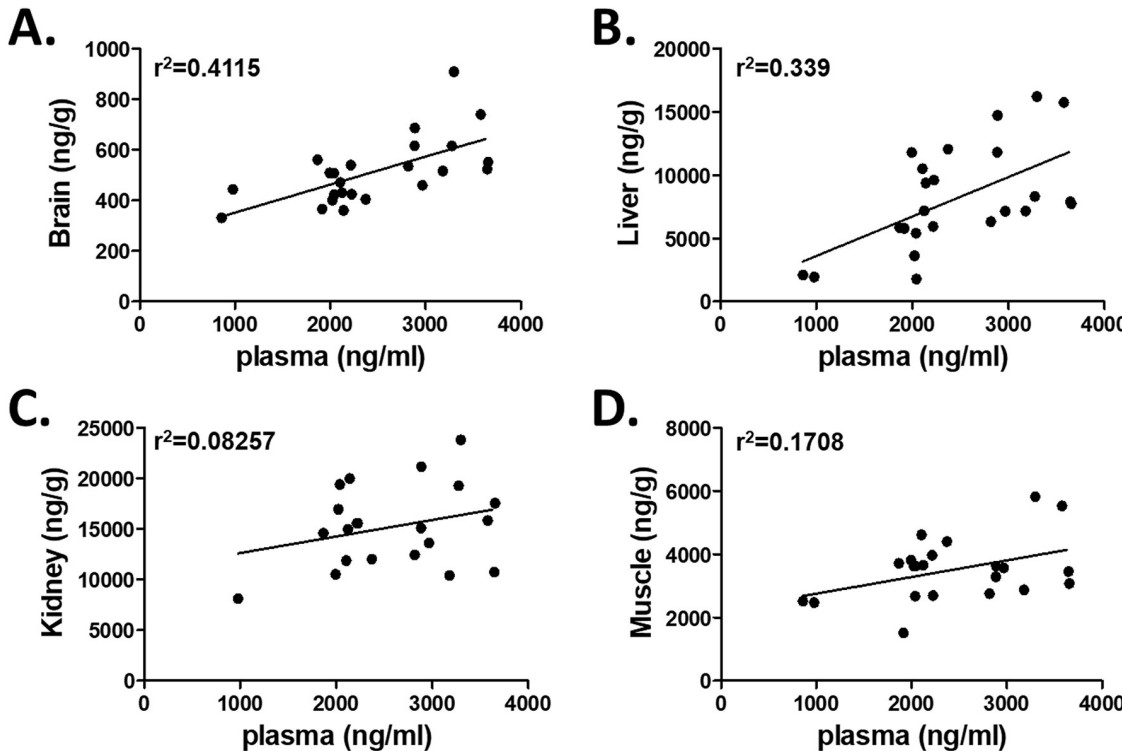

**Fig 4. Linear regression of metformin levels between plasma and different organs (n = 24).** Plasma and A. brain, B. liver, C. kidney, and D. muscle. Y-axis units are ng/g, and X-axis units are ng/ml.

doses of 500 mg metformin per 12 hours [35]. In the current study, steady plasma metformin concentrations were slightly but significantly higher in male mice than in females at 30 days after treatment. A gender difference of metformin accumulation was also observed in kidney, brain, and muscle. At 30 days after treatment, metformin concentrations in kidney, brain, and muscle were significantly higher in females than in males. A trend of higher metformin concentration in liver was also found in females than males although no statistical significance achieved.

Metformin mainly relies on OCTs for absorption and clearance. OCT2 is expressed in kidney which mediates renal clearance of metformin [36]. Interestingly, expression of OCT2 in kidney is gender-dependent with a much higher level in males than in females [37, 38]. Furthermore, testosterone has been found to increase while estrogen to decrease OCT2 expression in the kidney [38, 39]. We speculated that gender difference in OCTs expression might contribute to the gender difference of metformin concentrations between males and females. In addition, gastrointestinal transit tends to be slower in females and is subject to hormonal influence, which might also play a role in the gender difference of metformin pharmacokinetics [40].

Our metformin treatment paradigm in the mouse provides guidance for preclinical studies of metformin on aging and aging-related diseases. Treatment of metformin in drinking water has been commonly used in mouse studies at the concentrations of 0.5, 2, and 5 mg/ml or final dose of 50, 100, and 250 mg/kg body weight [41–51]. Treatment of 2 mg/ml metformin in drinking water has been found to increase survival time in male mouse model of Huntington's disease [41], increase motor unit survival in tibialis anterior muscles of female SOD1 amyotrophic lateral sclerosis mice without any significant effect on disease onset, progression or survival in male or female SOD1 ALS mice [48], increase generation of both intracellular and extracellular Aβ species [46], reduce tau phosphorylation but promote tau aggregation and exacerbates abnormal behavior in P301S human tau transgenic mice [49, 52], improve locomotor function while impair long-term memory in male C57B/L6 mice [50, 53]. Treatment of 5 mg/ml metformin in drinking water significantly reduced Aβ plaque burden and improved water maze performance in male and female APP/PS1 mice [51]. Treatment of 100 mg/kg/day metformin in drinking water has been found to increase mean and maximum life span of female transgenic HER-2/neu mice [42] and female SHR mice [43], decrease mean life span of male 129/Sv mice while increase mean life span female 129/Sv mice [45]. Treatment of 250 mg/kg/day metformin in drinking water provided most effective protection against metabolic disorder caused by a high carbohydrate-high fat diet in male CD mice as compared with 50 and 100 mg/kg/day metformin treatment [47]. Given the metformin dose range of 1,000 to 2,500 mg per day in humans, our study suggested that voluntary intake of metformin in drinking water at the concentrations of 2 to 4 mg/ml (final dose of 250 to 500 mg/kg/day body weight) will likely yield clinical relevant plasma concentrations of metformin in both male and female mice and that the studies using similar treatments could potentially lead to clinical translation.

## Comparison of current method with existing methods

Metformin is the most widely prescribed and marketed antidiabetic world-wide. It is not surprising that numerous methods for quantitation of metformin, or combinations existed (Table 4). However, the majority of published methods focused on plasma and few on other tissues such as liver and tumors. There are few papers discussing a robust method applied to different matrix such as brain. Some methods used complicated, error prone sample preparation (e.g. SPE, LLE), while other methods used strong concentrated buffer in LC-MS for good

**Table 4. Comparison of metformin assessment methods using LC-MS/MS.**

| Sr | Sample prep. | IS | RT (min) | Run time (min) | ion ratio | Column | Mobile Phase A | Matrix | Study | Method Validation | Reference |
|---|---|---|---|---|---|---|---|---|---|---|---|
| 1 | PPE[a], | Metformin-D6 | 0.89 | 3 | Yes | HR Xbridge C18, 50x3 mm,3.5 µm | 2 mM ammonium acetate | Mouse plasma, liver, kidney, muscle, brain, | correlation study | Yes | Current |
| 2 | PPE | Metformin-D6 | n/a | n/a | No | Aquasil C18, 2.1x20 mm, 5 µm | n/a | Mouse plasma, urine, liver, kidney, cage wash | Transporter effect to metformin tissue distribution | No | [54] |
| 3 | PPE | Metformin-D6 | n/a | 14 | Yes | Pursuit PFP, 2 x150 mm. 3 µm | 0.05% FA in water, 100% MPA-0% MPA gradient run Flow rate: N/A | Mouse Serum, Liver, tumor | dose study | No | [55] |
| 4 | PPE[b] | Alogliptin | 3.68 | 20 | Yes | Monolithic silica RP18, 100 x4.6 mm, 1.15 µm S | 10 mM ammonium formate pH3.0 80% MPA, 0.4 ml/min isocratic run | Human Plasma | PK | Yes | [56] |
| 5 | LLE | Glipizide | 1.84 | 6 | No | Peerless Basic C18, 33 × 4.6 mm, 5 µm | 0.5% FA in water, 20% MPA, 0.6 ml/min isocratic run | Human Plasma | Bioequivalence study | Yes | [57] |
| 6 | PPE | None | 0.98 | 14 | No | Gemini NX-$C_{18}$, 100 × 2.1 mm,3 µm | 10 mM ammonium acetate and 0.1% FA in water, 90%-5% MPA,0.25 ml/min Gradient run | postmortem blood | Matrix effect study | Yes | [58] |
| 7 | PPE | Diphen—hydramine | 2.6 | 3.5 | No | Zorbax SB C8, 150×4.6 mm 5 µm | 1.0% FA in water, 30% MPA, 0.5 ml/min, isocratic run | Human Plasma | PK | Yes | [23] |
| 8 | PPE | Moroxydine | 3.0 | 8 | No | Synergi POLAR-RP, 250 4.6 mm, 4 µm | 6 mM ammonium acetate and 0.1% FA, 50% MPA, 1.0 ml/min isocratic run | Dog plasma | PK | Yes | [59] |
| 9 | SPE | Metformin-D6 | 3.1 | 4.5 | No | XSelect HSS CN, 150 × 4.6 mm, 5 µm | 8 mM ammonium formate in water, pH4.5, 20% MPA, isocratic run | Rat Plasma | Bioavailability | Yes | [60] |
| 10 | PPE | Phenformin | 1.45 | 13 | No | Zorbax HILIC Plus, 50 × 2.5 mm, 3.5 µm | 0.1% FA in water, 25% MPA, 0.2 ml/min isocratic run | Rat plasma | PK | Yes | [61] |
| 11 | n/a | n/a | n/a | n/a | No | n/a | n/a | Human plasma and colonic tissue | correlation between plasma/colonic tissue | No | [62] |
| 12 | PPE-LLE | Propranolol | 1.0 | 5 | No | Zorbax C18, 50×4.6 mm, 5 µm | 0.1% FA in water, 60% MPA, 0.5 ml/min isocratic run | Human Plasma | PK | Yes | [63] |
| 13 | SPE | None | 3.5 | 5 | No | Cyano, 150 × 4.6 mm, 5 µm | 10 mM ammonium formate, 25% MPA, flow rate: N/A | Human Plasma | PK | Yes | [64] |
| 14 | PPE | Canagliflozin | 2.65 | 15 | No | Agilent Eclipse Plus C18, 4.6 × 100 mm, 3.5 µm | 6 mM ammonium formate and 0.1% FA, 0.5 ml/min, 98%—0% MPA, gradient run | Rat plasma | PK | Yes | [65] |
| 15 | PPE | Phenformin | n/a | n/a | No | Hypersil BDS C18, 150 mm × 2.1 mm,5 µm | n/a | Rat plasma, liver, kidney, intestine, muscle, heart | PK & PD | No | [66] |

*(Continued)*

**Table 4.** (Continued)

| Sr | Sample prep. | IS | RT (min) | Run time (min) | ion ratio | Column | Mobile Phase A | Matrix | Study | Method Validation | Reference |
|----|------|------|------|------|------|------|------|------|------|------|------|
| 16 | SPE | Alogliptin | 1.45 | 8 | No | Symmetry® C18, 4.6 x 50 mm, 5 µm | 10 mM ammonium formate and 0.2% FA, flow rate 0.25 ml/min, 5% MPA, isocratic run | Human Plasma | PK | Yes | [67] |
| 17 | PPE | Metformin-D6 | 2.2 | 3.5 | No | Kinetex HILIC, 50 x 4.6 mm, 2.7 µm | 9 mM ammonium formate, FA is water and 80 mM ammonium formate, FA in ACN | Human Plasma and urine | PK | Yes | [68] |
| 18 | PPE | N-despropyl ropinirole | 7.7 | 15 | no | Xbridge-HILIC BEH, 150x 2.1 mm, 3.5 um | 15 mM ammonium formate in water, 12% MPA, 0.25 ml/min isoractic run | Human Plasma | PK | Yes | [22] |
| 19 | PPE | Dapagliflozin | 0.78 | 6 | no | Acquity UPLC HSS Cyano, 100 x 2.1 mm, 1.8 µm | 0.1% FA in water, 90% -10% MPA, 0.4 ml/min, Gradient run | Rat plasma | PK | Yes | [69] |
| 20 | PPE | None | 1.2 | 9 | no | Xbridge C18, 2.1x50 mm, 3.5 µm | 1 mM ammonium formate with 0.1% FA, 95%-30% MPA, 0.2 ml/min gradient run. | Human plasma | PK | Yes | [70] |

[a] diluted with 2mM ammonium aceate;

[b] diluted with 10 mM ammonium formate;

PPE = Protein precipitation Extraction; LLE = Liquid-liquid Extraction; SPE = Solid Phase Extraction; FA = formic acid; PK = Pharmacokinetics study; n/a = Not available; RT = Retention time; MP = Mobile phase; IS = Internal standard.

chromatography. However, it was hard to solve low sensitivity and reproducibility issues. In majority of methods studying metformin in tissues, a long run time and other than metformin-D6 as internal standard could have lowered throughput without the ability to avoid matrix effect. Furthermore, none of the papers investigated the mechanism of metformin (biguanide) tautomerism feature, which influences chromatography in LC-MS/MS. The current method is sensitive and high throughput, but simple method to quantitate metformin in various tissues and plasma in a short total run time-3 minutes. Some of the strengths of our methods are as follows. Our method provides full validation covering plasma, kidney, muscle, liver, and brain. A simple and fast protein-precipitation method yielded a highly sensitive and reproducible method for multiple tissues as well as plasma. The use of metformin-D6 as an internal standard is more appropriate compared to previous papers, which fail to use labeled isotope metformin as an internal standard and relied on different chemical compounds with varied chemical properties than that of metformin. Further, we incorporated the ion ratio of both qualifier and quantifier to quantitate metformin in different tissues and plasma. Finally, our methods provide efficiency in the total sample run. The total run-time is only 3 minutes and post-run 1 minute, which minimized overall buffer usage and benefit the LC pump and MS source maintenance.

In summary, we have established a LC-MS/MS method for metformin measurement in plasma and different tissues. We found that a clinically relevant treatment paradigm of 4 mg/ml metformin in drinking in mice yielded plasma concentrations in the range of therapeutic

level in humans treated at the maximum dose of 2.5 g/day. Our metformin treatment paradigm in mouse and the established metformin levels in plasma, liver, muscle, kidney, and brain provide insight into future metformin preclinical studies for potential clinical translation.

## Supporting information

**S1 Fig. Representative calibration curves for tissues and plasma. A**. Plasma; **B**. Brain; **C**. Muscle; **D**. Liver; **E**. Kidney. The Y-axis shows the area of the peak (AUC) generated by the MS transition by LC-MS/MS. The X-axis shows the concentration analyzed. Data didn't deviate from linearity ($r^2$> 0.995) over the tested ranges. Data were fit by linear regression analysis.
(TIF)

**S1 Table. Metformin and metformin-D6 MRM parameters.**
(DOCX)

**S2 Table. Summary of linear regression from the calibration curve across plasma and tissues.**
(DOCX)

**S3 Table. Stability of metformin under different storage conditions.**
(DOCX)

## Author Contributions

**Conceptualization:** Kiran Chaudhari, Shao-Hua Yang.

**Data curation:** Kiran Chaudhari, Jianmei Wang, Yong Xu, Ali Winters, Linshu Wang.

**Formal analysis:** Kiran Chaudhari, Jianmei Wang.

**Funding acquisition:** Kiran Chaudhari, Shao-Hua Yang.

**Investigation:** Kiran Chaudhari, Jianmei Wang.

**Methodology:** Kiran Chaudhari, Jianmei Wang.

**Project administration:** Kiran Chaudhari, Ran Liu.

**Supervision:** Shao-Hua Yang.

**Validation:** Kiran Chaudhari, Jianmei Wang, Yong Xu, Xiaowei Dong, Eric Y. Cheng.

**Writing – original draft:** Kiran Chaudhari, Jianmei Wang.

**Writing – review & editing:** Shao-Hua Yang.

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
