## [Decision Letter · Decision Letter 0]

22 Apr 2020

PONE-D-20-08716

Determination of Metformin Bio-Distribution by LC-MS/MS in Mice Treated with a Clinically Relevant Paradigm

PLOS ONE

Dear Dr. Yang,

Thank you for submitting your manuscript to PLOS ONE. After careful consideration, we feel that it has merit but does not fully meet PLOS ONE’s publication criteria as it currently stands. Therefore, we invite you to submit a revised version of the manuscript that addresses the points raised during the review process.

Your manuscript was reviewed by two knowledgable referees in this area, and their comments are appended. As you will see they both recognize importance of your study, while they had several concerns that will need to be properly addressed by the authors before I can proceed further. In particular, the reviewer#2 raised a number of important technical points as well as manuscript itself. The authors need to address/respond to all their comments.

We would appreciate receiving your revised manuscript by Jun 06 2020 11:59PM. To enhance the reproducibility of your results, we recommend that if applicable you deposit your laboratory protocols in protocols.io, where a protocol can be assigned its own identifier (DOI) such that it can be cited independently in the future. For instructions see: http://journals.plos.org/plosone/s/submission-guidelines#loc-laboratory-protocols

We look forward to receiving your revised manuscript.

Kind regards,

Makoto Kanzaki, Ph.D.

Academic Editor

PLOS ONE

Journal Requirements:

2. At this time, we request that you  please report additional details in your Methods section regarding animal care, as per our editorial guidelines:

(a) Please state the number of mice used in the study  

(b) Please provide the name and dosage of the specific anaesthetic agent used during the cardiac puncture  

(c) Please clarify that the method of euthanasia used in the study was cardiac puncture

Thank you for your attention to these requests

Reviewers' comments:

Reviewer's Responses to Questions

**Comments to the Author**

1. Is the manuscript technically sound, and do the data support the conclusions?

Reviewer #1: Yes

Reviewer #2: Partly

2. Has the statistical analysis been performed appropriately and rigorously? 

Reviewer #1: I Don't Know

Reviewer #2: N/A

3. Have the authors made all data underlying the findings in their manuscript fully available?

Reviewer #1: Yes

Reviewer #2: No

4. Is the manuscript presented in an intelligible fashion and written in standard English?

Reviewer #1: Yes

Reviewer #2: Yes

5. Review Comments to the Author

Reviewer #1: Line No 175: mention the concentrations of QC samples.

Line No 190-194: As per the Bioanalytical Method Validation Guidance for Industry, precision and accuracy is recommended to perform at LLOQ, low, mid and high QC levels.

Line No 229-230: The author need to provide data for long term stability to justify the statement.

Reviewer #2: Comments on manuscript PONE-D-20-08716, Determination of Metformin Bio-Distribution by LC-MS/MS in Mice Treated with a Clinically Relevant Paradigm by Kiran Chaudhari, Jianmei Wang, Yong Xu, Ali Winters, Linshu Wang, Xiaowei Dong, Eric Y. Cheng, Ran Liu, and Shao-Hua Yang.

The authors developed a LC-MS method for the determination of metformin in mouse plasma, kidney, liver, muscle, and brain. The mice were voluntarily treated with metformin in drinking water. The study established steady-state metformin levels in the mentioned tissues and plasma. The topic is interesting and it fits the scope of the journal. However, there are many imperfections and they should be eliminated before a possible publication.

1) First of all, the authors declared in abstract “Nonetheless, very few metformin studies using mice have determined metformin concentrations…”. Despite the number of relevant reports is limited, there are published papers on the topic and some of them are focused specifically on metformin determination in mouse plasma and various tissues by LC-MS. Considering that one of the main goals of the manuscript is the demonstration of the new LC-MS method for metformin analysis in biologically relevant samples it is necessary to cite already published papers in introduction and to discuss them later in Results and discussion part. The authors should compare the existing methods with their newly developed procedure and show strong and weak features of the individual approaches. One table summarizing and comparing the main performance parameters of the methods would be very beneficial.

2) Page 5, line 100, Sample preparation and P6, L124, Preparation of calibration curve and quality control samples:

Here, it is unclear why the sample preparation procedure differs from the one used for calibration curve. Why amounts of plasma/tissue homogenates are not the same/similar in both cases? Moreover, I expect that metformin-D6 was added in the calibration samples, however, it is not mentioned in the text. Overall, the description of the procedure used for the preparation of calibration curve is incomplete, many other important parameters are missing.

3) P6, L112, Liquid chromatographic and instrumentation conditions:

The solvent gradient used (the time profile) is not described clearly. Moreover, 2 mmol/L ammonium acetate in water cannot be considered as a buffer. It is simply a salt of a weak acid and a weak base and buffering capacity of the resulting solution is very low. What is also surprising is the total run time 1.5 minutes, especially considering relatively low flow rate and, at the same time, the size of the column. Unfortunately, the authors did now specified the void time of the column, however, taking into account the size of the column, it is evident that practically no separation of analytes can take place in such short time and metformin is eluted together with other compounds very quickly and close to the void time of the column. Is there any time to re-equilibrate the column before the next injection? One can even ask for what there is a chromatographic column at all. The authors claimed later (in the validation part of the text) that due to the application of very selective MRM regime of MS and rigorous determination and measuremet of two MS transition ratio it was proved that the method was sufficiently selective. This might be the case. However, still, in my opinion it is appropriate to draw attention to the LC method being closer to FIA more than LC approch. In fact, when the authors try to explain the reason for the peak deformation (P7-8, L151-168 and Figure 2) by tautomerization of metformin, the real reason might be in the application of “unusual” chromatographic conditions on the column. Moreover, the authors’ explanation does not sound convinsingly furthermore when they claim the control of the pH by 2 mM ammonium acetate buffer, please, again consider that pH of such solution is not controlled at all. In fact, pH was much better controlled when they used 0.1 % HCOOH in H2O (and the peaks were distorted).

4) P8, L169, Method validation:

Matrix effect is discussed only and no information on recovery is provided. Moreover, the calculation of matrix effect is not described in sufficient detail (there is only a footnote below Table 2 and it is not clear what is meant by “analyte concentration in standard solution”). Thus, the matrix effect and recovery should be explained (procedure used for the calculation) and documented by data in adequate table.

5) P18, L349, References:

Official abbreviation should be used uniformly for publication names.

6. PLOS authors have the option to publish the peer review history of their article (what does this mean?). If published, this will include your full peer review and any attached files.

Reviewer #1: None

Reviewer #2: No

---

## [Author Response · Author response to Decision Letter 0]

18 May 2020

The response to editor's and reviewer's comments are provided as a separate document uploaded as "Response to Reviewer" word document.

---

## [Decision Letter · Decision Letter 1]

29 May 2020

Determination of metformin bio-distribution by LC-MS/MS in mice treated with a clinically relevant paradigm

PONE-D-20-08716R1

Dear Dr. Yang,

We are pleased to inform you that your manuscript has been judged scientifically suitable for publication and will be formally accepted for publication once it complies with all outstanding technical requirements.

With kind regards,

Makoto Kanzaki, Ph.D.

Academic Editor

PLOS ONE

Additional Editor Comments (optional):

Unfortunately, one reviewer could not review this revised manuscript. Based on my careful reading along with Reviewer#2's comments, I made this decision. 

Reviewers' comments:

Reviewer's Responses to Questions

**Comments to the Author**

1. If the authors have adequately addressed your comments raised in a previous round of review and you feel that this manuscript is now acceptable for publication, you may indicate that here to bypass the “Comments to the Author” section, enter your conflict of interest statement in the “Confidential to Editor” section, and submit your "Accept" recommendation.

Reviewer #2: All comments have been addressed

2. Is the manuscript technically sound, and do the data support the conclusions?

Reviewer #2: (No Response)

3. Has the statistical analysis been performed appropriately and rigorously? 

Reviewer #2: (No Response)

4. Have the authors made all data underlying the findings in their manuscript fully available?

Reviewer #2: (No Response)

5. Is the manuscript presented in an intelligible fashion and written in standard English?

Reviewer #2: (No Response)

6. Review Comments to the Author

Reviewer #2: (No Response)

7. PLOS authors have the option to publish the peer review history of their article (what does this mean?). If published, this will include your full peer review and any attached files.

Reviewer #2: No

---

## [Editor Report · Acceptance letter]

3 Jun 2020

PONE-D-20-08716R1 

Determination of metformin bio-distribution by LC-MS/MS in mice treated with a clinically relevant paradigm 

Dear Dr. Yang:

I'm pleased to inform you that your manuscript has been deemed suitable for publication in PLOS ONE. Congratulations! Your manuscript is now with our production department. 

Kind regards, 

on behalf of

Dr. Makoto Kanzaki 

Academic Editor

PLOS ONE